# Building better outcomes: A grounded theory approach to understanding creation and management of surgical data systems in Ethiopia

Taylor J. Jaraczewski[1]*, Tien Vo[2], Anika Agrawal[1], Haben Tafesse[3], Belay Mellese Abebe[3], Mary Schroeder[1], Christopher Dodgion[1], Katherine R. Iverson[1], Katinka Hooyer[4], Syed Nabeel Zafar[2]

1 Department of Surgery, Medical College of Wisconsin, Milwaukee, Wisconsin, United States of America, 2 Department of Surgery, University of Wisconsin-Madison School of Medicine and Public Health, Madison, Wisconsin, United States of America, 3 Department of Surgery, Hawassa University Comprehensive Specialized Hospital, Awassa, Ethiopia, 4 Department of Family and Community Medicine, Medical College of Wisconsin, Milwaukee, Wisconsin, United States of America

* taylorjarac@gmail.com

## Abstract

Surgical outcomes in low- and middle-income countries are significantly worse than in their high-income country counter parts. Registries are an important tool in quality improvement as they facilitate collection and analysis of comprehensive patient data to be used for benchmarking outcomes. Despite the known efficacy of such registries, they are sparse in low- and middle-income countries. Ethiopia, however has had success implementing perioperative data collection systems at a national as well as local level. This qualitative study seeks to understand the experience of creating and managing such registries in the low-income setting from the perspective of individuals who work in this space. We performed 14 semi-structured interviews with individuals from Ethiopia who work with data systems. Interviews were performed with 3 data collectors, 3 surgeons, 3 administrators, 2 hospital focal people, and 3 ministry of health officials. Thematic analysis was employed through a grounded theory approach. Primary themes were identified: the resource ecosystem, the budget checkpoint, digitalization, the mindset highway, and universal personal commitments. From these themes, we developed an interdependent systems theory, which states that complex data systems in LMICs are driven by the dynamic interaction of three pillars: infrastructure, people, and budget prioritization. This study provides rare insight into the management of peri-operative data collection systems in LMICs from the perspective of professionals who work in this space. Findings from this study can be utilized as a reference and blueprint for other countries and groups who aim to create data collection systems in challenging environments.

**Data availability statement:** Interview transcripts utilized for this study can be found at https://doi.org/10.7910/DVN/RDR9QA. All transcripts have been redacted to ensure interviewee confidentiality.

**Funding:** This work was funded by the Association for Academic Surgery Foundation Trainee Global Surgery Research Fellowship Award, awarded to TJJ. The funders had no role in study design, data collection and analysis, decision to publish, or preparation of the manuscript.

**Competing interests:** The authors have declared that no competing interests exist.

## Introduction

Surgical outcomes in low- and middle-income countries (LMICs) are disproportionately poor compared to high-income countries (HICs) [1]. The GlobalSurg collaborative, a network of over 5,000 clinicians in 100 countries who conduct studies on international surgical outcomes, published three cross-sectional cohort studies each showing that mortality was several fold worse in LMICs for emergency abdominal and cancer surgeries compared to HICs [2,3]. Data driven quality improvement (QI) programs have proven to be effective to improve peri-operative outcomes in HICs [4]. However, the availability of patient data tends to be a significant hurdle for many LMIC-based health systems.

Many HIC and LMICs have utilized focused data sets to capture specific sets of health metrics [5]. This is in line with the World Health Organization (WHO) recommendation that baseline assessment and ongoing surveillance plans are the first step for surgical system strengthening [6]. In 2016, Ethiopia was one of the first sub-Saharan African countries to devise a plan that included a system for monitoring and evaluation of surgical care at a national level. This initiative, called Saving Lives Through Safe Surgery (SaLTS), included a pillar to collect seven surgical indicators integrated into an open-source health information management system called District Health Information Software 2 (DHIS2) [7,8].

In January 2021 our group, comprised of US-based surgeons and residents from the Medical College of Wisconsin (MCW) and University of Wisconsin-Madison (UWM), partnered with the department of surgery at Hawassa University comprehensive specialized hospital (HUCSH) in Hawassa, Ethiopia to design and implement a REDCap based comprehensive perioperative registry that collects preoperative, intraoperative, postoperative, and 30-day outcomes data [9]. The registry was formally started in May 2022 and has successfully collected data on over 1,000 surgical cases. It has been integral for a department wide research-oriented culture shift, increased QI educational training, and a substantial increase in surgical outcome research by both residents and faculty.

In this study we aimed to bring to light the distinctive experience of collecting and managing perioperative data in the LMIC setting through interviews with key partners who have engaged in this space in Ethiopia. By dissecting the complexity of the situation from different perspectives (data collectors, clinicians, hospital administrators, and Ministry of Health (MOH) personnel) we hoped to better understand the data collection and management process in the LMIC setting and utilize these experiences to inform expansion efforts of current registries and implementation of new registries in these settings.

## Methods

### Design

A qualitative research design was chosen to understand social interactions, processes, and cultural dynamics associated with barriers, facilitators, and perceptions of collection and management of perioperative data in Ethiopia, an LMIC. This design allowed for a deeper understanding of context dependent phenomena currently under-researched.

## Participants and setting

Ethiopia was selected because of prior partnerships between the research team and interview participants in the country. Further, through programs such as the Ethiopian Hospital Alliance for Quality (EHAQ) and Saving Lives through Safe Surgery (SaLTS), the country has made significant efforts to expand data systems and improve quality [10,11]. A purposive sampling technique was utilized focusing on participants who the research team knew had extensive experience with data systems in some capacity, whether that be through direct data capture, data management, and/or data storage and utilization. Recruitment was performed between October 1, 2023 – January 30, 2024, which resulted in a total of 14 subjects. Written and verbal consent were obtained. To ensure a diverse constellation of perspectives we interviewed data collectors (n = 3), surgeons (n = 3), administrators (n = 5), and MOH officials (n = 3, Table 1). Data collectors were made up of general practitioners and one general surgery resident. General practitioners are individuals who have completed medical school and a one-year internship and are utilized at the discretion of the departments for whom they work. Surgeons were all individuals who were clinically active, had experience in data systems, and held leadership positions within their department or hospital. Administrators were comprised of individuals in hospital leadership, quality office leadership, key performance indicator (KPI) focal persons who collect targeted data for the national health management information system (HMIS), and one individual who leads a private organization that collects data at a national level. Given the small niche sample size, quotations are presented without attribution to protect participant anonymity.

## Semi structured interviews

An interview guide was generated using the Performance of Routine Information System Management (PRISM) framework (S1 File) [12]. This framework states that routine health information performances are affected by the processes, which result from technical, behavioral, and organizational determinants. The PRISM framework has been extensively and successfully utilized in the LMIC setting to study data collection for diseases such as HIV and hepatitis C [13,14]. The guide was tailored to each individual's role. Questions were focused on 3 primary areas: 1) Perspectives and attitudes 2) Barriers to collection and management 3) Facilitators of collection and management. All interviews were conducted by

**Table 1. General demographics of interview subjects.**

|  | Gender | Role |
|---|---|---|
| *Data collector 1* | F | General Practitioner |
| *Data collector 2* | M | Resident |
| *Data collector 3* | M | General Practitioner |
| *Surgeon 1* | M | Chief of surgery |
| *Surgeon 2* | M | Chief of surgery |
| *Surgeon 3* | M | CEO, Surgeon |
| *Administrator 1* | M | Data collection expert, PO |
| *Administrator 2* | M | Medical Director |
| *Administrator 3* | M | Chief of Quality |
| *KPI focal person 1* | M | HMIS employee |
| *KPI focal person 2* | M | Focal officer |
| *MOH 1* | M | MOH employee |
| *MOH 2* | M | MOH employee |
| *MOH 3* | M | MOH employee |

M = Male; F = Female; GP = General Practitioner; CEO = Chief executive officer; PO = private organization; HMIS = Health management information system; MOH = Ministry of health.

one member of the research team (TJ). Given the semi structured format of the interview, we utilized a laddering interview technique, as described by Bourne *et al* [15]. Briefly, key topics were explored utilizing broad questions to identify important attributes from the perspective of the interviewee. Subsequent rounds of probing questions, based on the identified attributes, were then utilized to explore more deeply these attributes at increasing levels of abstraction. The interview guide was not formally vetted by our international partners as it was intended as a flexible framework rather than a fixed instrument, which is consistent with the grounded theory approach. The laddering approach facilitated participants to guide the content and depth of discussion. As such, the majority of the data generation occurred through probing based on participant lead priorities, rather than through strict adherence to pre-specified questions. One interview was conducted virtually, while the rest were in person. Every interview was recorded and transcribed with Otter.ai®. Each interview lasted 20–35 minutes. All recordings were reviewed and edited by an independent trained transcriber as well as the interviewer. Interviews were not intended to reach thematic saturation but instead focused on generating sufficient depth and richness of data to meaningfully address the research question at hand.

### Thematic analysis and validation

Thematic analysis was performed through a grounded theory approach with a combination of inductive and deductive coding, as highlighted by Strauss [16,17]. Grounded theory was chosen as it allows for theory to emerge directly from stakeholders' experience. Guided by the PRISM framework, deductive codes included behavioral, technological, and organizational barriers and facilitators. All other topics that did not fit within these categories, were coded inductively. Coding was performed by one individual (TJ), the interviewer. Analysis was carried out using MAXQDA Analytics Pro (version 24.3.0).

To ensure rigor and high-fidelity data, a staged approach was utilized. Open coding was conducted alongside the writing of memos to capture contextual nuances and to develop interconnections between codes and quotations. After coding all transcripts, codes were collapsed as appropriate. Axial coding was then employed to group codes into refined categories. Finally, definitions for each theme were thoughtfully crafted to accurately reflect the categories and coded segments.

Validation was performed using multiple techniques. Triangulation of the themes was performed using member-checking with 3 of the interviewees as well as review by other members of the research team who have experience with data systems in Ethiopia. Member checks were also used throughout the interviews by restating or summarizing to ensure the interviewer had correct understanding. Memos were utilized in a reflexive manner to ensure that researcher biases were acknowledged. A PhD level expert in qualitative methods reviewed each step of the process to ensure methodological rigor. Included within this was a code audit on 3 of the interviews to ensure coding consistency, reliability and trustworthiness. Finally, a sampling method of individuals with different roles added diverse perspectives.

### Ethics

IRB approval was received from HUCSH (Reference number IRB/327/15) and an exemption under category 45 CFR 46: (2)(ii) Tests, surveys, interviews, or observation (low risk) was granted from the University of Wisconsin-Madison (ID number 2023–0842). Human subject research activities inadvertently occurred while this project had MCW approval under 45 CFR 46.118, which was disclosed to the MCW IRB who determined the data did not have to be destroyed and could proceed with publication. All participants were provided both written and verbal consent.

### Results

Five main themes emerged: the resource ecosystem, the budget checkpoint, digitalization, the mindset highway, and universal personal commitment (Table 2). Each theme had role-specific nuances. Themes are supported below with raw data. The first four themes were manifest (more literal) whereas the fifth, personal commitment, was latent (more interpretative).

**Table 2. Summary of developed themes and definitions of each theme.**

| Theme | Definition |
|---|---|
| *Theme 1: The resource ecosystem* | An interdependent network of resources that synergistically support and enhance data collection and management processes |
| *Theme 2: The budget checkpoint* | The unique stop/go effect that finances hold throughout every aspect of the data collection and management continuum |
| *Theme 3: Digitalization* | The mentality that all facets of data collection and management must be adapted to be operated with the use of computers |
| *Theme 4: The mindset highway* | A delicate state of cognitive cohesion in regards to data collection, management, and use |
| *Theme 5: Universal Personal Commitment* | Determination to implement and optimize data capture and management systems |

## Theme 1: The resource ecosystem

The resource ecosystem is the interdependent network of resources that synergistically support and enhance data collection and management processes. The lack of any resource within the system will lead it to crumble. This network includes human, technological, financial, raw materials, and training resources. As one participant noted:

> "So it's not an easy task [to collect and manage data], it needs collaboration of different stakeholders, and not only stakeholders contribute to the data quality, other infrastructures, technologies, different resources are also required to collect quality data. "

The prioritization of each resource was dependent on the role of the interviewer (Fig 1). For data collectors, shortages of basic materials were the most pressing concern. As one explained:

> "One of the things we are facing currently is just we don't have that much of paper. So there have been some shortage. And that was a problem. There was just some gap, shortage of paper."

This quote shows the lack of paper being a significant hurdle, but this is not the only resource. Printer ink, internet, and appropriate tools to collect data were all resources that, when lacking, led to issues.

Administrators and surgeons also discussed the importance of materials and technology such as poor internet access as a significant barrier. However, they included an additional topic in the form of the human resources (i.e., people to collect data):

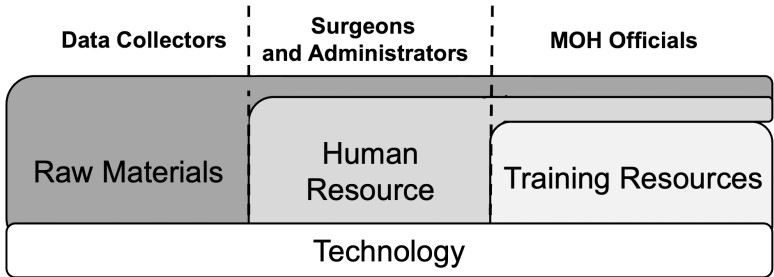

**Fig 1. Types of resources emphasized by role.** Resources were repetitively brought up as important throughout the transcripts. However, the resources emphasized were different depending on the role of the interviewee. All interviewees placed a heavy emphasis on technology. Data collectors focused more on raw materials such as paper and data collection tools. Surgeons and administrators placed a larger emphasis on the availability of people to actually collect data. MOH officials mentioned both of these prior resources, but also added in the importance of resources for training.

*"Data collector is a challenge. Data collection by itself is a challenge. So there is no separate data collectors.. which is assigned for these outcomes. So we are using our faculty members, staff members, certified department staff members. So for me it creates some challenges like arranging these, creating arrangements, using residents and so on."*

Their perspectives highlight how the absence of designated data staff can divert clinical personnel from patient care and subsequently strain the hospital operation.

While all previous resources were mentioned, MOH employees added in the concept of resources for training and how to work to facilitate a top-down approach to providing resources to the hospitals throughout the country. One official described how programs like EHAQ were designed to provide financial support, mentorship, and training to large hospitals that could then cascade knowledge and resources to surrounding facilities:

*"Yeah, actually, if you take what we call the, the EHAQ, this alliance… As a ministry, we cannot be all over the country. So [we] just select.. strong hospitals.. And then we select those hospitals and we give finance, we give the mentorship, we give some training, and then they will do the rest with their cluster hospitals. So actually, if you take the St. Paul, it will have some, like four or five hospitals inside [it]. So we give… the technical, the finance, we do it for the St. Paul and they will do [for those other hospitals]."*

Taken together, these accounts illustrate how the resource ecosystem encompasses an intricate network of human, technological, financial, and material resources that are essential for effective data management. The absence or insufficiency of any of these resources was repeatedly highlighted as significant barriers that hindered the data collection process.

## Theme 2: The budget checkpoint

The budget checkpoint can be defined as the unique stop/go effect that finances hold throughout every aspect of the data collection and management continuum. Whether it was digitalization, hiring data collectors, acquiring resources or performing research finances emerged as the central "checkpoint" that influenced every stage of data collection and management. Participants repeatedly emphasized that even when there was consensus on the value of interventions, progress is often halted by limited budget. As one administrator noted:

*"Actually, there is no other reason rather than budget. Budget. Yeah, it takes a lot of budget."*

The sentiment was echoed across roles, with interviewees underscoring that financial resources were the ultimate deciding factor in whether systems could be developed or sustained. For example, another participant elaborated on how digitalization was stalled despite widespread agreement:

*"So, we are trying to digitalize but there are some financial constraints. We agree on the digitalization, we start to digitalizing, but it didn't finish because of the financial restraint."*

The budget checkpoint overlapped with the resource ecosystem as it was seen as the ultimate and most critical resource (Fig 2). While interviewees that worked at local hospitals (data collectors, surgeons, and administrators) mentioned how it was difficult to acquire funds, MOH officials discussed the dichotomy of implementing systems nationwide while constantly being asked to provide financial support:

*"So most of hospitals are complaining that we don't have budget. [They say] you design the strategy, but you don't have the budget, you don't give us the budget. That is one of the longest standing complaints from the hospitals."*

**Fig 2. Relationship between resource ecosystem and budget checkpoint.** The budget checkpoint acts as the most important node within the ecosystem. Acquiring high quality data is not feasible without appropriate budget.

Budget constraints served as a pivotal checkpoint influencing every stage of data collection and management, often overriding consensus on desired interventions. These accounts illustrate how the budget checkpoint overlaps with the broader resource ecosystem but stands out as one of the most critical factors. Across all levels, from local hospitals to the MOH, financial limitations emerged as the critical determinant of implementation success.

### Theme 3: Digitalization

The digitalization theme is defined as the mentality that all facets of data collection and management must be adapted to be operated with the use of computers. This was shared by interviewees in every role. One participant captured this sentiment directly by saying:

> *"The system should be developed to collect the data, one it has to be digital, the computer-based information system should be in place."*

Many transcripts mentioned how currently utilized digital technologies have greatly enhanced data collection and management, compared to paper charts and manual data extraction. For example, in this excerpt one of the participant discusses how data capture has been made significantly easier with the software REDCap, a web-based and mobile phone-based platform designed to support HIPAA-compliant data capture [18]:

> *"I think I will continue with this REDCap because I found it easier to fill or to put data there, at least for data entry, [it does] not need internet connection. Just mobile phone is enough for entry. You do not need to go in another ways of inputting data like sophisticated ways of data entry, like computers are much big thing and even you don't need to have hard copies, loss of papers… and it can even be filled by one person, so many charts."*

Yet it was evident that not all hospitals in the country were at the same stage of digitalization. One participant described how technological advances are not evenly distributed around the country:

*"As you know, Ethiopia is a low-middle income country. So…some technical technological gadgets are not distributed, as expected… So [acquiring] the real time data is challenging in some rural areas."*

Others noted that while select hospitals have begun to use electronic medical records (EMR), nationally these were quite rare. An anecdote from one participant who works at one of the few hospitals in the country that has EMR reflected on the importance of expanding such systems throughout the country:

*"One thing is that, for example, we have the hospital who start this EMR two years back and make it use digital medical recording system. So we get data whenever we want in the system… I'm really proud that physicians [are] starting to publish in my university…We are working to integrate so all… the public health specialists will learn clinical scenarios… and the physicians and nurses will learn how to make analysis and interpretation of that."*

Digitalization was not relegated to only collecting and managing data, but also for displaying data to help hospitals and governments monitor metrics. A participant elaborated on this:

*"We are helping the hospitals to monitor the data that they have by providing them regular dashboards and data entry software, which is also helping them to monitor their data... and also interpret the data that they have. So I think this is also helping a lot of people. We are also closely working with academic institution[s]… to use it for their students, research activity and… conducting different community services."*

Overall, digitalization emerged as a universal goal, with participants emphasizing the necessity of computer-based systems for efficient data collection, management, and interpretation. While technologies like REDCap and EMRs have streamlined processes and supported research, varying levels of technological adoption across hospitals highlight the ongoing challenge of achieving nationwide digital integration.

### Theme 4: The mindset highway

Beyond material and technological needs, participants stressed the importance of the stakeholder mindset. The mindset highway is conceptualized as a delicate state of cognitive cohesion in regards to data among all stakeholders. As one person stated:

*"First thing is engagement of stakeholders. That is crucial. The leaders, the professionals, and even the political leaders, to engage them in why surgical data is important, important down the line, in the primary hospitals to the national level. That's critical."*

This state is highly vulnerable to disruptions either in the form of individuals veering off of the mindset or not getting on the highway at all. One of the critical components of ensuring an effective data system was ensuring that all stakeholders understood the importance of data.

One anecdote in reference to the archive staff who work in the chart room suggested a lack of understanding on their part, and the necessity to explain the significance of data collection:

*"Most of them… have not that much educational qualification to understand what we are doing. They're more of labor workers. So, they know that people in hospitals, especially University Hospital, are doing research… but we have to try to explain what we were doing [and how it will] benefit the hospital."*

However, this knowledge gap was not unique to the archive staff. As one participant pointed out, many professionals have poor attitudes and/or knowledge gaps, which can lead to a lack of quality data collection as well:

*"The first thing is, as I told you, it is a wrong attitude because they, they think that [data collection is a] secondary duty, not a primary duty for them, especially in clinical setting. All the surgeons, nurses, and anesthetists, they just want to practice... They don't want to recollect [data]."*

Not only did the lack of knowledge about or will to collect data cause issues, but also a lack of understanding on how the data will be utilized. Many transcripts mention a fear of retribution when it comes to poor outcomes; thus, many people will purposely not collect high fidelity data.

*"If they give you a real data, they feel that they will… be incriminated... So they don't want to share that. The major challenges I think, for example, there are some guys… that sometimes just give the ideal data rather than what's happening, so we tend to reject that data because we know they are falsifying it."*

To ensure that people stay in the correct mindset and do not veer off track many interviewees described monitoring systems set in place. At the local quality office level as well as MOH level, training and monitoring strategies were utilized to ensure stakeholders provide high quality data. One person explained:

*"We have regular mentorship and supportive supervision. All of the wards, we mentor the staff, data owners, head nurses, we mentor regularly…The HMIS officer should be always alert. Alert to mentor and supervise the staff and always it should be continuous process."*

Further, members from the MOH discussed the implementation of a number of different programs to help facilitate collaboration and a unified mindset. EHAQ was one such program, which was mentioned previously. One individual also elaborated on the SALTs program:

*"I think one of the biggest things that we've done so far is we try to develop the strategy. So I think that there is a strategy, the SALTs 2, saving lives through safe surgery... a few see that strategy [as trying to] address all the issues, even including data related issues. So one of the things that, [you do is] you develop strategy, and you orient that strategy for the hospitals to adhere with that."*

Many interviewees discussed that the mindset towards data collection should be reflected by incorporating data collection into the clinician's job. However, this should not be an added task, but instead merged with current activities. One participant gave one solution to this dilemma:

*"If the person who is really close with that patient, who is doing the procedure or something like that, can fill the data [while] they are writing charts…then it may make things easier."*

Participants also elaborated on the effect of stakeholders not being unified when it comes to data collection. They report that this discordance can lead to significant barriers downstream. For example, one person discussed how a lack of understanding of the importance of data can lead to a lack of funding:

*"The understanding [of] the importance of data is not there…. And because of that… there is a financial constraint, a budget constraint, they give priority for other [initiatives]. So, surely not only the budget constraint, but the of gap of understanding."*

Ultimately, the mindset highway reflects the necessity of fostering a unified cognitive alignment among stakeholders regarding the importance of data. Gaps in understanding or poor attitudes towards data collection were identified as key barriers, necessitating strategies like training, mentorship, and integration of data tasks into existing workflows to ensure sustained engagement and high-quality data practices.

**Theme 5: Universal personal commitment**

Despite the barriers described in earlier themes, participants consistently expressed determination and optimism for improving data systems. This theme reflects the resilience and determination of participants in striving to implement and optimize data capture and management systems. One person stated:

*"It's not 100% perfect, but like I have out there, in time that our effectiveness and efficiency has grown up in time. So we are, these things require progress, and they require time to develop. So it's progressing, like we are, every time we have a meeting and decide some things to do then. We have been progressing and almost we are reaching to be perfect in course of time... We have progressed a lot."*

Throughout all of the transcripts a constant latent theme of determination was palpable. Even when discussing challenges, barriers, and failures, interviewees would often end with a sense of optimism. Negative situations were often viewed as learning points, instead of impassible barriers. One person emphasized ongoing problem-solving:

*"But different issues need to be discussed and.. addressed in terms of data security, in terms of data storage, in terms of data management, and all these kinds of need…So we are trying to address these kinds of issues. And I can say we have addressed most of them. But there are still issues that we need to address. And the more we [partner with others], the more we get other institutions who are working on improving data systems.. [it] will get closer to perfection."*

Universal personal commitment embodies the resilience, determination, and optimism of participants in overcoming challenges to improve data systems. Despite setbacks, interviewees consistently framed obstacles as opportunities for growth, demonstrating a collective resolve to progress toward more effective and efficient data management. This theme emanates through all of the previous themes as many interviewees offered next steps and future directions, which are highlighted in the previous sections.

## Discussion

In this study interviews were conducted with key stakeholders in Ethiopia who had significant experience with data collection and management in the LMIC setting. Subjects represented a wide range of roles, from on-the-ground data collectors to MOH officials responsible for overseeing national data collection efforts. Five themes were identified including the resource ecosystem, the budget checkpoint, digitalization, the mindset highway, and universal personal commitment. While all themes were present throughout the transcripts, many varied in emphasis depending on the individual's role.

### Emergent themes

LMICs have been described as suffering from the inverse care law – "the mortality that existing effective healthcare technologies could eliminate if they were delivered successfully to all those who can benefit." [19]. We extend this to include data systems. Merely supplying resources is insufficient. Instead, their collective interactions determine impact [20]. The resource ecosystem reflects the idea that individual resources themselves are not as impactful as their collective presence. Like a biological ecosystem, the absence of one component can destabilize the whole. An intertwining nature exists between resources including human, material, financial, and technological, that must be maintained to ensure success.

One interviewee provided an account that described this multi-dimensional system, including an acute shortage of human resources for data collection and management. The limited number of available personnel were often overworked, which adversely affected the quality of their output. Moreover, due to inadequate compensation, these individuals often had to undertake additional employment or reduce working hours to manage familial responsibilities. Reports of poor quality eventually reached leadership and administration and, ultimately, disincentivized future financial investments.

Among all resources, finances act as the system's gateway. Throughout the interview transcripts, individuals in every role repeatedly emphasized that everything comes down to budget. Despite their detrimental impact, financial constraints in LMICs have been shown to stimulate frugal innovation in LMICs, fostering adaptive and resource-efficient solutions [21]. One example from our study was EHAQ, a national multiorganizational QI collaborative that cascades MOH training from referral hospitals to regional [22]. While this has had success, budget constraints still persist and have impeded the training of smaller hospitals. Additionally, even when training is conducted, the expectations for data collection and reporting through the DHIS2 system are often unachievable due to the budgets.

Digitalization offers a strategic solution to overcome budget constraints in data collection by minimizing resource demands. One study demonstrated that digitalized data capture reduced research costs by 17–62% compared with paper-based methods in both HIC and LMIC settings [23]. By embedding data collection into routine charting, digitalization could streamline processes, decrease needed materials, and reduce reliance on additional staff [24].

The Ethiopian MOH is currently working to implement wide spread application and use of electronic medical record (EMR) systems around the country. While this has been gradual, primarily due to reluctance on the side of clinicians to learn a digital system, efforts for nation-wide adoption of EMR continues [25]. This uptake will not only improve efficiency of patient care but also improve data completeness up to 40% and speed up to 20% compared to paper records [26]. A longstanding criticism of current EMRs in HICs is that while they are effective for documentation, they were not created efficiently for research or QI purposes [27]. While EMR incorporation in LMICs is still in early stages, creating them in a manner that facilitates data collection for QI is one area that could help rapidly accelerate QI work in the future.

The mindset highway theme was a concept that permeated throughout the pages of every transcript. While every interviewee had a different flavor to this theme all of them followed a very similar pattern: all stakeholders must be on the same page, and all members of the medical system must view data collection and QI as one of the primary objectives in their job. If this objective was treated as secondary or as not part of their job, the system will crumble.

The analogy of a highway provides a multifaced comparison that exemplifies the dynamics of data collection and management systems. The concept of lanes, where all individuals travel in a unified direction, highlights the importance of alignment and attitude towards data. Divergence from these lanes leads to chaos and ultimately inhibits progress. The MOH has attempted to promote unity in attitude towards data by providing standardized sets of surgical metrics through the SaLTS program [28]. The ultimate ramifications of this program, and if it leads to alignment with the Lancet commission on global surgery goals for 2030 are still yet to be seen [29].

Another aspect of this analogy is the idea of continuous process improvement. On a highway there are painted lines and rumble strips to guide or correct the course of any vehicle that strays. This parallels well with the experience of the HMIS and MOH interviewees who discuss ongoing mentorship and quality assessments. The process that they discussed is akin to a continuous QI process comprised of repetitive feedback loops.

The passion for data acquisition was palpable throughout all of the interviews. This was particularly evident when participants discussed challenges and ongoing issues. It was rare that any interviewee merely identified a problem; instead, they often, unprompted, proposed solutions that they were either currently implementing or planning to implement. Among the interviewees there was a recognition of the potential improvements that data collection could have on their health system, specifically in regards to surgical access and quality. Though it should be noted that such optimism could be a reflection of Ethiopian culture, and may be different in other settings.

PLOS Global Public Health

## The interdependent systems theory

Through this comprehensive grounded theory process, a core overarching theory emerged that we have termed the interdependent systems theory. This theory posits that the development and sustainability of data systems in LMICs depends on the interplay between three pillars: budget, infrastructure, and people. These function as interdependent systems that collectively shape the functionality of data systems across geographic and institutional contexts. Any weakness in any of these pillars leads to the data system crumbling while coordination enables sustainability. Fig 3 depicts a conceptualization of this theory. The infrastructure portion includes both innovation and resources, thus encapsulating both themes 1 and 3. It underscores the necessity for robust materials and resources to be available for any type of registry to be successful. However, this should be accompanied by a culture that fosters improvement and innovation, such as digitalization. The system of people reflects the 'mindset highway' and 'universal personal commitment.' The budget pillar highlights the importance of a sustainable financial commitment. It stands as a singular entity to signify its place as a checkpoint, as highlighted in theme 2.

For those seeking to establish a data system in an LMIC, this theory offers a foundational framework for sustainability and a clear starting point. Initial steps should include a thorough evaluation of infrastructure and resources, alongside an assessment of individual partner and organizational engagement and commitment. Equally, if not more critical is aligning budgetary decisions with strategic goals to ensure long-term feasibility. Additionally, implementing an auditing system to regularly assess and optimize each pillar of the framework ensures they function cohesively and effectively.

The interdependent systems theory builds upon previous implementation frameworks, such as the PRISM framework, by providing a more targeted set of domains focused on data systems in LMICs. Our theory presents three interdependent pillars that directly emerged from the grounded data. These findings not only simplify the application of the theory, but also highlight budget as a discrete checkpoint for long-term sustainability. Further, by including innovation (e.g., digitalization) within infrastructure and elevating human commitment as a defining pillar, our theory underscores the dynamic interconnectedness of context, culture, and resources. In doing this, the interdependent systems theory both complements and

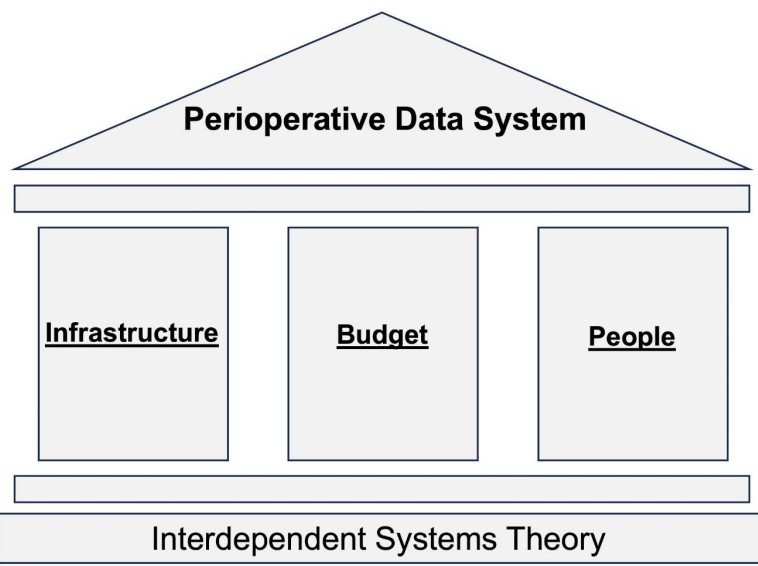

**Fig 3. The interdependent systems theory.** This theory proposes that the creation and sustainability of perioperative data systems is dependent on a proper balance of infrastructure, people, and budget.

extends upon existing frameworks, offering a novel lens that is tailored to the realities of data system development in resource-limited settings.

## Limitations

This unique qualitative analysis on data systems in LMICs should be understood in the context of its limitations. Given the design and intention, it is not generalizable. Every country and population have different cultures, which can significantly impact the results. However, we believe the themes and concepts are broad and will resonate in most resource limited settings. Second, one interviewer did all the interviews and had at least a partial acquaintance with all interviewees. This familiarity could have an influence on how the interviewees answered questions. However, it is hard to know whether this would lead to more or less transparency. Formal intercoder reliability procedures were not employed, as coding was conducted by a single researcher. This represents a potential source of interpretive bias. To enhance rigor and trustworthiness, coding and theme development were conducted under the supervision of a PhD level qualitative methods expert to help decrease the risk of bias or preformed ideas by having iterative analytical meetings and formal coding audits.

## Conclusion

Ultimately, the continued growth of surgical data collection and management systems within the LMIC setting is critical to achieve safe surgical care around the world. Such systems will serve as a rich resource for conducting extensive studies on surgical outcomes, understanding current states, and developing interventions to improve the quality of surgical care in LMICs. This study identified multiple factors necessary for sustainability: 1) the interdependent role of budget, infrastructure, and people in shaping data systems, 2) the potential of digitalization to streamline processes and improve data quality and 3) importance of a unified mindset among stakeholders. These findings provide a practical framework for groups and countries seeking to initiate or strengthen surgical data systems.

## Supporting information

**S1 File. Semi structured interview guide.**
(DOCX)

**S1 Checklist. Inclusivity in global research.**
(DOCX)

## Author contributions

**Conceptualization:** Taylor J. Jaraczewski, Christopher Dodgion, Katherine R. Iverson, Katinka Hooyer, Syed Nabeel Zafar.

**Data curation:** Taylor J. Jaraczewski, Tien Vo, Anika Agrawal, Haben Tafesse, Belay Mellese Abebe, Katherine R. Iverson, Syed Nabeel Zafar.

**Formal analysis:** Taylor J. Jaraczewski, Katinka Hooyer.

**Funding acquisition:** Taylor J. Jaraczewski, Mary Schroeder.

**Investigation:** Taylor J. Jaraczewski, Katinka Hooyer.

**Methodology:** Taylor J. Jaraczewski, Christopher Dodgion, Katinka Hooyer.

**Project administration:** Tien Vo, Anika Agrawal, Belay Mellese Abebe, Mary Schroeder, Christopher Dodgion, Katinka Hooyer, Syed Nabeel Zafar.

**Resources:** Haben Tafesse, Belay Mellese Abebe, Mary Schroeder, Katherine R. Iverson, Katinka Hooyer, Syed Nabeel Zafar.

**Software:** Taylor J. Jaraczewski, Katinka Hooyer, Syed Nabeel Zafar.

**Supervision:** Mary Schroeder, Christopher Dodgion, Katherine R. Iverson, Katinka Hooyer, Syed Nabeel Zafar.

**Validation:** Tien Vo, Anika Agrawal, Haben Tafesse, Belay Mellese Abebe, Katinka Hooyer.

**Visualization:** Taylor J. Jaraczewski.

**Writing – original draft:** Taylor J. Jaraczewski.

**Writing – review & editing:** Taylor J. Jaraczewski, Tien Vo, Anika Agrawal, Haben Tafesse, Belay Mellese Abebe, Mary Schroeder, Christopher Dodgion, Katherine R Iverson, Katinka Hooyer, Syed Nabeel Zafar.

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
