## [Decision Letter · Decision Letter 0]

28 Jul 2025

PGPH-D-25-01668

Building better outcomes: A grounded theory approach to understanding creation and management of surgical data systems in Ethiopia

Dear Dr. Jaraczewski,

Thank you for submitting your manuscript to PLOS Global Public Health. After careful consideration, we feel that it has merit but does not fully meet PLOS Global Public Health’s publication criteria as it currently stands. Therefore, we invite you to submit a revised version of the manuscript that addresses the points raised during the review process.

We look forward to receiving your revised manuscript.

Kind regards,

Barnabas Tobi Alayande

Academic Editor

Journal Requirements:

Additional Editor Comments (if provided):

Reviewers' comments:

Reviewer's Responses to Questions

**Comments to the Author**

1. Does this manuscript meet PLOS Global Public Health’s publication criteria?

Reviewer #1: Yes

Reviewer #2: Yes

2. Has the statistical analysis been performed appropriately and rigorously?

Reviewer #1: N/A

Reviewer #2: N/A

3. Have the authors made all data underlying the findings in their manuscript fully available (please refer to the Data Availability Statement at the start of the manuscript PDF file)?

Reviewer #1: Yes

Reviewer #2: Yes

4. Is the manuscript presented in an intelligible fashion and written in standard English?

Reviewer #1: Yes

Reviewer #2: Yes

Reviewer #1: This is a qualitative study from Ethiopia examining the factors that lead to successful implementation of surgical data registries.

1. There is a typo in the second paragraph of the introduction. The sentence should read "In 2016, Ethiopia was on the first sub-Saharan African countries" - sub-Saharan should be hyphenated and country should be countries.

2. In the same paragraph, probably District Health Information System should be capitalized.

3. In the methods, the authors should include a paragraph about how the data registry was set up. What year was it developed? What was the process for setting up the registry? What data is collected? Who enters the data? Is this person(s) paid for this job? How many people total are involved in maintaining the database? If so, how is it funded? Did the data collectors receive specialized training? On what kind of data system is the data stored? Who has access to the data? Who owns the data? Etc.

4. Likewise, it is not clear from the methods and table 1 if all of the people interviewed are directly involved in data collection? Or if you are also seeking opinions from those more peripherally involved.

5. The interview guide should be included as a an appendix. Also, was the interview guide vetted with your international partners?

6. In the paper, I would consider removing interview subject labels from specific quotes. In many LMICs, there are limited people in certain roles and by assigning, for example "MOH1" to a certain quote, this person's anonymity may not be protected. I would just keep the quotes but leave the labels out.

7. The section that discusses the themes (starting at Theme 1: The resource ecosystem") needs to be written in a more fluid way. It seems very choppy. The quotations are just inserted, without a paragraph explaining the context or the relevance to the overall findings. There is also no indication of how many (or what percentage) of interviewees in the study agreed with the sentiment being discussed. Also, the paragraph that begins Fig 2. types of resources emphasized by role - is this a figure legend? Or is it part of the text? This section about the themes needs to be completely rewritten to be smoother and tell a story to the reader about what you have found from the research. This is the case for the entire section Themes 1-5. This entire section needs to be rewritten.

8. The discussion section is a bit unorganized. It should be broken into pieces with subtitles highlighting the different topics, and should also be written more concisely (remove a lot of words).

9. Figure 1 should be removed. It does not add anything to the paper, as the methods are straightforward and easily understood without the figure.

10. The discussion should probably spend more time talking about the framework that was used and how the data can fit into that.

11. The conclusion needs to be rewritten. It is a very vague conclusion, which does not explain the main findings of the study. The conclusion should tell the reader, what are the 2-4 things that this study found?

Reviewer #2: General comment

The manuscript is a valuable and timely contribution to addressing persistent data collection challenges in the surgical space. This thoughtful assessment arrives at a critical juncture in Ethiopia’s transition toward digitalization of health care records and identifies critical barriers faced by both surgical data systems and broader healthcare information infrastructure. Appreciation goes to the authors for engaging in such a meaningful and necessary inquiry. Their work offers insight as the country moves toward more efficient data-driven healthcare delivery. However, the paper could benefit from addressing the following comments.

Abstract

There is a mention of interview participant roles twice. To improve clarity, you might first state the total number of interviews, then list participant types before describing the analysis method. This would streamline the abstract and avoid repetition.

Introduction

In the first sentence of the introduction, the term many countries is vague. It's better to specify which countries, HIC, LMICs? Does focused data here refer to surgical data or any health metrics? I believe the paragraph can be better introduced with better specificity.

The authors on the last paragraph of the introduction stated “.. by dissecting the complexity of the situation from different perspectives…”. The different perspectives can be specified here to attract the reader's attention … clinicians, data collectors to MOH personnel etc.

Methodology

The methods section lacks clarity, unlike the abstract section that states both thematic analysis and grounded theory were used in combination. I recommend stating this clearly in the methodology section and clarifying and justifying the use of grounded theory in the research.

The manuscript does not explain how the final sample size of 14 interviews was determined, nor whether thematic saturation was achieved. Including a brief statement on saturation and sampling adequacy would strengthen methodological transparency.

The manuscript also does not mention any intercoder reliability process. Given the interpretive nature of the fifth (latent) theme and the policy implications of the findings, it's important to indicate how the authors maintained rigor in this aspect. If a single person coded all transcripts it's better to mention lack of intercoder reliability in the limitation section.

Results

Under theme 1, in the first paragraph after the first excerpt, the sentence “ The prioritization of each resource was dependent on the role of the interviewer…” appears to have typo error as the role pertains to to the interviewee and not the the interviewer.

After the third excerpt under theme 1, the paragraph describing a “top down” approach is likely referring to an adaptive mechanism used by MOH for the shortage of resources for training. The authors can be more explicit in their description here for readers' clarity.

Under theme 2, in the last paragraph, it would be helpful to include a few more interview excerpts showing how budget constraints overrode stakeholder consensus on preferred interventions. This would make the theme clearer and more convincing.

Theme 3_ In the statement “one of the only hospitals in the country that has EMR,” the use of “only” may unintentionally imply exclusivity, suggesting there is a single such hospital. To enhance clarity and accuracy, it's better to replace it with “one of the few hospitals”, which more precisely conveys the limited, but not singular availability of EMR systems in Ethiopia.

Theme 4_ The term day laborers working in the chart room can be misleading as it might be referring to janitors, cleaners or guards, who have no role related to data or chart keeping. The phrase in the excerpt from interviewee “They're more of labor workers” might be referring to the inadequacy of their knowledge by comparing them to labor workers. The Authors, however, should be careful in using the term "day laborers" to describe archive staff. This is repeated 3 times under these theme four and should be corrected or the authors should present more substantial evidence of “day laborers” in the chart archive rooms.

Discussion

The discussion section is written in a well elaborated and attractive manner. However, there are several claims that need citation and proper substantiation of claims. Please see the following as an example.

The statement that “financial constraints are a main driver of innovation in LMICs” lacks a supporting citation. Moreover, it appears within an argument that otherwise frames finance as an obstacle. Recommend rephrasing to reflect the paradoxical nature of constraint-driven innovation and including references from frugal innovation literature to strengthen the claim.

On paragraph 4 of the discussion section, the authors mention digitalization as a strategic solution to budget constraints. Several important claims are made about the role of digitalization. These would be stronger if substantiated with concrete examples from literature showing how digital tools simplified complex processes or materials in practice. Likewise, the statement that digitalization could “facilitate broader participation in the data acquisition system, in a non-intrusive manner” would benefit from specific instances from literature or interview excerpts where digital systems demonstrably aided inclusive, low-disruption data collection.

The statement referencing a 50% initial capture rate and subsequent deployment of a registry manager provides valuable insight, but it is presented only in the Discussion. The authors should provide a clear context for this discussion point in the result section.

The attribution of participant optimism to “Ethiopian culture” is interpretive yet unsupported. To enhance scientific credibility, suggest either framing this as a speculative hypothesis or providing sociocultural references that substantiate this interpretation.

Finally, Interdependent systems theory, the framework developed through infrastructure, people, and budget prioritization is attractive and clearly grounded in the identified themes. However, it may benefit from a deeper articulation of how it innovates beyond existing frameworks like PRISM. The authors could clearly state this as their contribution to existing literature.

**Do you want your identity to be public for this peer review?** For information about this choice, including consent withdrawal, please see our Privacy Policy

Reviewer #1: No

Reviewer #2: No

---

## [Decision Letter · Decision Letter 1]

11 Dec 2025

PGPH-D-25-01668R1

Building better outcomes: A grounded theory approach to understanding creation and management of surgical data systems in Ethiopia

Dear Dr. Jaraczewski,

Thank you for submitting your manuscript to PLOS Global Public Health. After careful consideration, we feel that it has merit but does not fully meet PLOS Global Public Health’s publication criteria as it currently stands. Therefore, we invite you to submit a revised version of the manuscript that addresses the points raised during the review process.

Thank you for improving this manuscript significantly based on reviewer comments. Please apply very minor revisions to proceed, thank you.

Document clearly that the qualitative interviews were not intended to reach saturation, based on the arguments you present so that your work is not misrepresented or misunderstood.

Because you only dealt with 14 individuals and limited individuals in say the MoH, they can be easily identified using the quote attributions as proxy identifiers. The authors' rebuttal to the reviewer's suggestion is reasonable, however, it might be of interest to the participants not to be identified by those adjacent to the study. Please leave out the attributions with the quotes. This is commonly done with small, niche sample sizes. As a protective measure, if you would like to include them, kindly include consent for interviewees to be represented in this current potentially identifying manner.

Also kindly include reflections on why the interview guide was not vetted by international partners, as part of reflexivity.

In addition, please respond to the current reviewer's comments to proceed.

We look forward to receiving your revised manuscript.

Kind regards,

Barnabas Tobi Alayande

Academic Editor

Journal Requirements:

Reviewers' comments:

Reviewer's Responses to Questions

**Comments to the Author**

Reviewer #2: All comments have been addressed

publication criteria?

Reviewer #2: Yes

3. Has the statistical analysis been performed appropriately and rigorously?

Reviewer #2: Yes

4. Have the authors made all data underlying the findings in their manuscript fully available (please refer to the Data Availability Statement at the start of the manuscript PDF file)?

Reviewer #2: Yes

5. Is the manuscript presented in an intelligible fashion and written in standard English?

Reviewer #2: Yes

Reviewer #2: The Authors have addressed nearly all comments provided. Thank you for the attention given to the feedbacks. Please revisit the following two comments that still need correction in the manuscript.

Not addressed: “one of the only hospitals in the country that has EMR,” the use of “only” may unintentionally imply exclusivity, suggesting there is a single such hospital. To enhance clarity and accuracy, it's better to replace it with “one of the few hospitals”, which more precisely conveys the limited, but not singular availability of EMR systems in Ethiopia. Not Addressed

While the authors note that a code audit was conducted on three interviews and that diverse sampling contributed to the richness of perspectives, these steps do not fully substitute for formal intercoder reliability procedures. Because coding appears to have been conducted primarily by a single researcher, the absence of multiple independent coders should be acknowledged as a methodological limitation, along with its implications for interpretive bias.

**Do you want your identity to be public for this peer review?** For information about this choice, including consent withdrawal, please see our Privacy Policy

Reviewer #2: No

---

## [Editor Report · Decision Letter 2]

15 Jan 2026

Building better outcomes: A grounded theory approach to understanding creation and management of surgical data systems in Ethiopia

PGPH-D-25-01668R2

Dear Dr. Jaraczewski,

We are pleased to inform you that your manuscript 'Building better outcomes: A grounded theory approach to understanding creation and management of surgical data systems in Ethiopia' has been provisionally accepted for publication in PLOS Global Public Health.

Best regards,

Barnabas Tobi Alayande

Academic Editor

All comments have been appropriately addressed.